# TRAINING DEEP NEURAL NETWORKS BY OPTIMIZING OVER NONLOCAL PATHS IN HYPERPARAMETER SPACE

## ABSTRACT

Hyperparameter optimization is both a practical issue and an interesting theoretical problem in training of deep architectures. Despite many recent advances the most commonly used methods almost universally involve training multiple and decoupled copies of the model, in effect sampling the hyperparameter space. We show that at a negligible additional computational cost, results can be improved by sampling *nonlocal paths* instead of points in hyperparameter space. To this end we interpret hyperparameters as controlling the level of correlated noise in training, which can be mapped to an effective temperature. The usually independent instances of the model are coupled and allowed to exchange their hyperparameters throughout the training using the well established parallel tempering technique of statistical physics. Each simulation corresponds then to a unique path, or history, in the joint hyperparameter/model-parameter space. We provide empirical tests of our method, in particular for dropout and learning rate optimization. We observed faster training and improved resistance to overfitting and showed a systematic decrease in the absolute validation error, improving over benchmark results.

## 1 INTRODUCTION

The remarkable improvement in performance of machine learning models has been paid for with an increased model complexity. Part of this complexity is directly related to architectural choices, such as model topology and depth or increased overall number of parameters (weights), but an increasingly important aspect, from both theoretical and practical standpoint, is the growth of the number of *hyperparameters*. Though the classification may seem fluid, with structural features such as the number of layers being treated as a hyperparameters as well, the difference is laid bare by the distinct nature of training, which involves a double loop of optimizations. The inner loop finds the best model weights – with hyperparameters held fixed to some arbitrary values – using the performance on the training dataset, while the outer loop is responsible for finding the optimal value of hyperparameters to ensure a robust performance on new data, i.e. generalization. A separate dataset is used for this key task. Hyperparameters thus optimized may include structural features, regularization constants, and even dynamical characteristics of the training algorithm.

The practical importance of hyperparameters cannot be underestimated: the very same architecture may deliver state-of-the-art or mediocre results, depending on the choices made. Simultaneously as the number of hyperparameters grows, the search space becomes high-dimensional and thus necessitates proper hyperparameter optimization (HO) procedures, which go beyond rule-of-thumb choices. This is additionally complicated by the fact that computational resources and time are usually limited. A number of HO approaches have been developed (Bergstra et al., 2011); the simplest ones involve grid- or random searches, but their main weakness is poor scaling and failure to reuse information obtained from previous parameter choices. A broad family of methods addressing the latter issue arise in the Bayesian framework: here the the assignment of new hyperparameter values to be evaluated is informed by the performance of the model with the previous assignments (Snoek et al., 2012). The high computational cost of evaluation inspires additional strategies, for instance bandit methods, to leverage cheaper but cruder results obtained with partial datasets. All of the above HO strategies, however, share a key characteristic: optimization of hyperparameters is decoupled from that of the weights, and the procedures amount to a more or less clever sampling of the points in hyperparameter space, using the validation dataset only. A family of genetic HO algorithms overcome this issue by optimization in both weights and hyperparameter space, however, in

this case, in a greedy way (see for example Jaderberg et al. (2017)). They cannot be thus guaranteed to explore space in an unbiased way, and the results may explicitly depend on initial conditions .

Here, we propose a radically different strategy: that of coupling the usually independent copies of the model at different hyperparameter values *during* the training. The instances of the model are allowed to exchange their hyperparametes as the optimization of the weights on the training dataset progresses, in effect tracing out a *highly nonlocal* trajectory in the product of hyperparameter and weight spaces. The final trained model cannot thus be thought of as being characterized by a particular point value of the hyperparameters, but rather by a path or history. The exchange procedure is based on the physical technique of parallel tempering (Swendsen & Wang, 1986), which itself depends on a mapping of the hyperparameter values to an effective "temperature" of the model. We demonstrate empirically, that this HO approach – which leverages also the training dataset – has a number of desirable properties: it results in models more resilient to overfitting, achieving smaller overall errors, and achieving them faster. The method is naturally parallelizable and the computational cost overhead over usual grid searches is negligible.

The goal of this work is then twofold: first, to provide (or, in some instances, recapitulate) a unifying and intuitive physical perspective on various types of hyperparameters, which can be interpreted as setting the level of "noise" during the training of the model. This, in turn, defines an effective parameter - similar in spirit to temperature - controlling the smoothness of the generalization function. Second, and much more important in applications, is to show that this perspective allows to move beyond the usual paradigm of two decoupled weight/hyperparameter optimizations in a well-motivated and controlled fashion, and ultimately to train more robust models.

This paper is organized as follows: in section 2 we review the interpretation of hyperparameters as controls of noise, and the mapping to an effective "temperature". In section 3 we introduce the main concepts of our path-based model optimization. In section 4 we show numerical tests of the approach on neural nets trained on EMNIST and CIFAR-10 datasets. The discussed examples of hyperparameters include the learning rate and drop-out rate, among others. Finally, in section 5 we discuss the implications, as well as potential generalizations of the method.

## 2 HYPERPARAMETERS AS CONTROLS OF SMOOTHNESS OF POTENTIAL LANDSCAPE

Training of modern machine learning models, in particular of deep neural networks (DNNs), is a complex non-convex optimisation problem of minimizing the total objective function (otherwise called loss or energy):

$$\mathcal{L}(\mathbf{x}, \mathbf{W}) = \frac{1}{N} \sum_{i=1}^{N} l_i(\mathbf{x}_i, \mathbf{W}), \tag{1}$$

where $l_i(\mathbf{x}_i, \mathbf{W})$ captures deviations of the predictions of the model, parametrized by weights $\mathbf{W}$, for the $i$-th data point $\mathbf{x_i}$ in the training set of total size $N$. The standard choice of the optimisation method for this problem remains the Stochastic Gradient Descent (SGD) algorithm

$$\mathbf{W}_{t+1} = \mathbf{W}_t - \frac{\gamma}{|N_B|} \sum_{i \in N_B} \nabla_{\mathbf{W}} l_i(\mathbf{x}_i, \mathbf{W_t}) \tag{2}$$

with a step size $\gamma$ and the gradient computed only on a batch $N_B$ of the training set.

It is well established, that various forms of direct noise injection, whether to the inputs, weights or gradients, aid generalization performance (Sietsma & Dow, 1991; Neelakantan et al., 2015). From the theory point of view it has been shown that the effect of such methods is analogous – though not identical – to introducing particular types of regularization (Bishop, 1995; An, 1996) smoothing the cost function. Furthermore, an often invoked intuition is that noise allows the algorithm to escape narrow minima of the cost function defined by the training data, thereby preventing overfitting. This is also true for non-white noise introduced "indirectly", such as the one generated by the mini-batches in Eq.2. Aided by the results suggesting poor generalization for "deep" (Choromanska et al., 2015), and, conversely, good generalization for "flat" minima of the loss function (Hinton & van Camp, 1993; Hochreiter & Schmidhuber, 1995), it was postulated in Zhang et al. (2018) that strong performance of SGD is, in fact, due to the correlated nature of the induced batch noise.

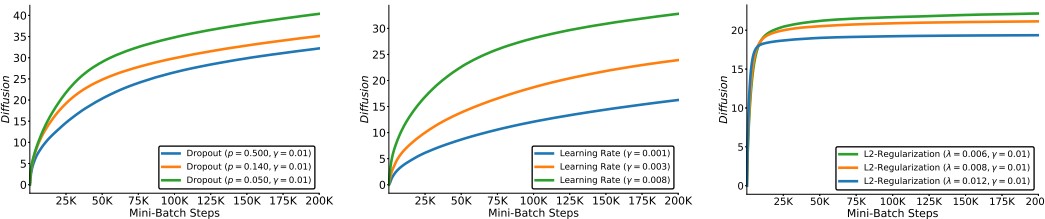

**Figure 1:** Diffusion curves for model weights (calculated as Euclidean distance between weights vector at initial time $t_0$ and the vector at epoch $t$) for replicas of the model at different rate of Dropout, learning rate and L2 regularization. See discussion in the main text for details.

In what follows, we will greatly benefit from an intuitive – if not always rigorous – picture relating various kinds of noise to an effective parameter controlling smoothness of the potential landscape, similar in spirit to temperature. Common hyperparameters, such as the learning rate, dropout or batch size, can be treated on the same footing, as we now demonstrate.

Let us begin with the prototypical case of added Langevin noise, for which the temperature analogy is exact (Seung et al., 1992): the gradients are corrupted by white noise $\xi$ with a variance $\langle \xi_i(t)\xi_j(t') \rangle = 2T\delta_{ij}\delta_{tt'}$. The naive GD can then be viewed as a diffusion process of particle in a complex potential landscape, with the weight updates a discretized version of the continuum equation:

$$\frac{\partial \mathbf{W}}{\partial t} = -\nabla_{\mathbf{W}}\mathcal{L} + \xi(t). \tag{3}$$

At long-times Eq. 3 converges to a Gibbs probability distribution, from which weights $\mathbf{W}$ can be directly sampled:

$$P(\mathbf{W}) = \frac{1}{Z}\exp\left(-\beta\mathcal{L}\left(\mathbf{W}\right)\right). \tag{4}$$

The inverse temperature $\beta = 1/T$ is not arbitrary, but rather is defined by the variance of the noise $\xi$. Furthermore, the multiplicative prefactor $\beta$ in Eq. 4 controls the overall scale of variation of the potential landscape. Strong noise, corresponding to large variance, and therefore high temperature T, results in small $\beta$ which "flattens" the landscape of $\mathcal{L}\left(\mathbf{W}\right)$. Conversely, weak noise results in large $\beta$, amplifying potential differences. This is the central idea behind changing temperature in the familiar simulated annealing method (Kirkpatrick et al., 1983). An alternative way of phrasing the above is that the increased variance of the noise does not allow the finer details the potential landscape to be explored by the dynamics, effectively smoothing it. The crucial observation is that the latter perspective extends also to cases where the noise is not white. Though, strictly speaking, the variance of the noise is not the temperature anymore, it still performs an analogous function: it effectively controls the roughness/smoothness of the landscape. An immediate consequence is that higher variance facilitates diffusion of the model in such high-dimensional space, and improves ergodicity (see Fig. 1).

As an illustration, we now consider common examples of hyperparameters from this perspective, and examine their influence on diffusion curves. For the learning rate $\gamma$, its magnitude is directly related to the size of the time step in the discretization of the Langevin equation, and by dimensional analysis it is analogous to increasing the noise variance (thus temperature) by a factor of $\gamma$. In the case of SGD dynamics Eq. 2 the noise variance can be computed explicitly (Zhang et al., 2018). Even though the noise is highly correlated, i.e. not thermal, the variance scales overall as inverse batch size: $|N_b|^{-1}$, and thus smaller batch size smoothens the potential. Similarly, dropout regularization has the effect of increasing the variance of neural outputs at training by a factor of inverse dropout retention rate $p^{-1}$ (Hinton et al., 2012), therefore amplifying noise and improving diffusion, as seen in Fig. 1. The case of L2 regularization is different: an increased L2 naturally diminishes the magnitude of the potential landscape variations, and so the initial diffusion is faster (see Fig. 1), however at later times the diffusion is effectively suppressed; the plateau value reached is, naturally, the lower the stronger the L2 regularization. Since, therefore, the models do not diffuse at different rates, L2 does not satisfy the basic requirements of our procedure, and we do not expect systematic improvements for this hyperparameter. On the other hand, for parameters relating to direct noise injection (for instance to gradients or weights), the relationship of large magnitude to

high "temperature" is natural, as described above, and their smoothing effect has been noted in the literature An (1996).

Since generically there are many hyperparameters, and results or intuitions such as above may not be readily available, it is imperative to be able to systematically identify whether they are related to landscape smoothness. This is, fortunately, possible. The functional significance of the notion of smoothness lies precisely in determining the diffusion properties of the random walk, therefore it is natural to test for smoothing properties by performing a *diffusion experiment*. To wit, sensitivity of the weight diffusion curves – obtained from SGD training runs – to the value of hyperparameter in question implies relation to smoothness. To exemplify this we consider Batch Normalization (BN). By studying Lipschitz properties Santurkar et al. (2018) ultimately confirmed it is related to effective landscape smoothing. Alternatively, comparing training with and without BN, it can be immediately shown to improve weight diffusion (see Fig.2).

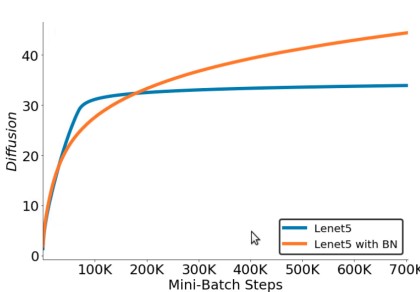

**Figure 2:** Weight diffusion curves with and without Batch Normalization(BN)

Training the model with different hyperparameters can thus be intuitively thought of as minimizing the objective at different intensities of noise, or, equivalently, in effective landscapes of varying degree of smoothness. We will now use this insight to construct a new nonlocal hyperparameter optimization procedure.

## 3 REPLICA EXCHANGE OF HYPERPARAMETERS

---

**Algorithm 1** Training with replica exchange

---

**INPUT:** Number of replicas $M$, Inverse "temperature" (hyperparameters) $\beta = (\beta_1, \beta_2, \ldots, \beta_M)$
        Number of steps for initialization $\Delta N_i$
        Number of SGD steps between exchanges $\Delta N_e$
        Exchange normalization parameter $C$
        Number of steps $T$
**OUTPUT:** Weight configurations $\mathbf{W} = (\mathbf{W_1}, \mathbf{W_2}, \ldots, \mathbf{W_M})$ of the replicas,
  1: **Initialization:** $\forall k \in M$, initialize weights $\mathbf{W}_k$ for each replica and set $t = 0$.
  2: $\forall k \in M$, perform SGD for $\Delta N_i$ steps. Update $t \leftarrow t + 1$ at each step.
  3: **Repeat:**
  4:     $\forall k \in M$, perform SGD for $\Delta N_e$ steps to update $\mathbf{W}_k$. Set $t \leftarrow t + 1$ at each step.
  5:     Let $\mathcal{L}_t = (\mathcal{L}(\mathbf{W}_1^t), \mathcal{L}(\mathbf{W}_2^t), ..., \mathcal{L}(\mathbf{W}_k^t))$ be validation losses at time $t$.
  6:     Randomly select a pair $(m, n)$ of replicas with adjacent temperatures.
  7:     **if** $\Delta = C(\beta_m - \beta_n)[\mathcal{L}(\mathbf{W}_m) - \mathcal{L}(\mathbf{W}_n)] \leq 0$ **then**
  8:         swap $\beta_m$ and $\beta_n$
  9:     **else**
10:         swap $\beta_m$ and $\beta_n$ with probability $\exp(-\Delta)$.
11:     Finish if $t > T$.

---

The core of our approach is to endow the model with the ability to change hyperparameters during training, and hence to optimize the model over a subset of paths in the combined weight and hyperparameter space, as opposed to being confined to hyperplanes determined by a fixed value of the hyperparameter. In the previous section we have noted that varying typical hyperparameters defines a family of optimization problems of monotonically changing smoothness of the effective landscape, and thus of ergodicity properties and complexity. This family is labelled by an effective "temperature" determined by the variance of the noise induced in the training by particular hyperparameter choices.

Inspired by similar problems in statistical physics we use the parallel tempering (replica exchange) method (Swendsen & Wang, 1986). In this procedure, multiple Markov Chain Monte-Carlo

(MCMC) simulations, or replicas, are run in parallel at different temperatures, which define the levels of uncertainty in the objective function, i.e. the energy. The temperatures are arranged in ascending manner forming a ladder on which the states of the replicas are swapped between neighboring values with Metropolis-Hastings acceptance criteria, ensuring that the whole setup satisfies detailed balance not only for each chain individually, but also between the chains (see discussion below). The lower temperature chains' ergodicity is thus radically improved by the possibility of temporarily performing MC moves at higher temperatures, where the landscape looks flatter. Note that this results in a *non-greedy* and *highly nonlocal* exploration. Indeed, parallel tempering is efficient in systems with broken ergodicity – the usual case for complex energy landscapes, where configuration space can effectively become partitioned into separate regions with low probability for inter-region transitions (for a concise review see Earl & Deem (2005)). Applications beyond the original spin-glass simulations include protein folding (Fukunishi et al., 2002) and, in machine learning, training of Boltzmann machines (Desjardins et al., 2010).

In our algorithm, the MCMC chains for individual replicas are replaced with multiple instances of the standard SGD dynamics distinguished by hyperparameter values, which is a more efficient way to approach Langevin-like equation 3 for systems with large number of long-range correlated parameters. We assume, using the noise analogy discussed in the previous section, that the weight configurations $\mathbf{W}$ of the replicas follow a Gibbs distribution $P_m(\mathbf{W})$ characterized by the effective inverse "temperature" $\beta_m$, determined by the value of the hyperparameter:

$$P_m(\mathbf{W}) = \frac{1}{Z_m} \exp\left(-\beta_m \mathcal{L}\left(\mathbf{W}\right)\right), \tag{5}$$

with $Z_m$ the partition function. The total system distribution is then a product over the $M$ replicas:

$$P_{\text{full}} = \prod_i^M P_i(\mathbf{W}_i) \tag{6}$$

The transition probability $\mathcal{P}\left(\mathbf{W}, \beta_m; \mathbf{W}', \beta_n\right)$ of swapping configurations $\mathbf{W}'$ and $\mathbf{W}$ between two replicas $m$ and $n$ should satisfy the detailed balance condition in the full system, which implies:

$$\frac{\mathcal{P}\left(\mathbf{W}, \beta_m; \mathbf{W}', \beta_n\right)}{\mathcal{P}\left(\mathbf{W}', \beta_n; \mathbf{W}, \beta_m\right)} = \exp\left(-\Delta\right), \tag{7}$$

where we used Gibbsianity and where:

$$\Delta = (\beta_m - \beta_n)\left[\mathcal{L}\left(\mathbf{W}'\right) - \mathcal{L}\left(\mathbf{W}\right)\right] \tag{8}$$

Note that the global detailed balance condition guarantees that in the long-time limit the joint probability distribution Eq. 6 will be sampled faithfully. In particular, no dependence on initial state of the weights will survive. Following the standard Metropolis-Hastings scheme, the exchange occurs with analytically computable acceptance probability:

$$\begin{aligned}\mathcal{P}\left(\mathbf{W}, \beta_m; \mathbf{W}', \beta_n\right) &= 1 \quad \text{for} \quad \Delta \leq 0, \\ &= \exp\left(-\Delta\right) \quad \text{for} \quad \Delta > 0.\end{aligned} \tag{9}$$

The resulting Algorithm 1 is described in the table above. It accepts $M$ replicas of the system, a vector $\beta$ of inverse temperatures, number of initialization and SGD steps between exchanges. Each one of the $M$ realisations of the system is first run for $\Delta N_e$ steps, until they achieve their relative equilibrium. Subsequently, exchanges are proposed every $\Delta N_e$ steps. The constant $C$ introduced in the acceptance ratio is responsible for normalization of the exponential argument and may be required when the values of hyper-parameter are very low, or too high (see below).

An important practical issue is the selection of replica temperatures and spacing, so that exchange proposals are accepted with a high probability. The theoretical answer depends on the properties of the energy landscape and fluctuations in the system; the basic idea is to ensure that energy histograms for neighbouring replicas have sufficiently large overlap. To this end, a common choice for temperature ladder is a geometric progression. In what follows, for simplicity, we use an equal spacing between replicas and instead scale $\beta$ with a tunable parameter $C$ to control the exchange rate, which requires calibration. For such tuning, running of preliminary realisations of the system prior to a parallel tempering optimisation may be needed to approximate energy histograms. Here we run short training without exchanges to tune the acceptance ratio constant $C$, or to fine-tune the temperatures selection. It is also worth noting that the acceptance ratio relates to the number of trainable parameters in the model. Increase in the number of parameters allows for more accessible states in parameter space, and induces smaller acceptance ratio.

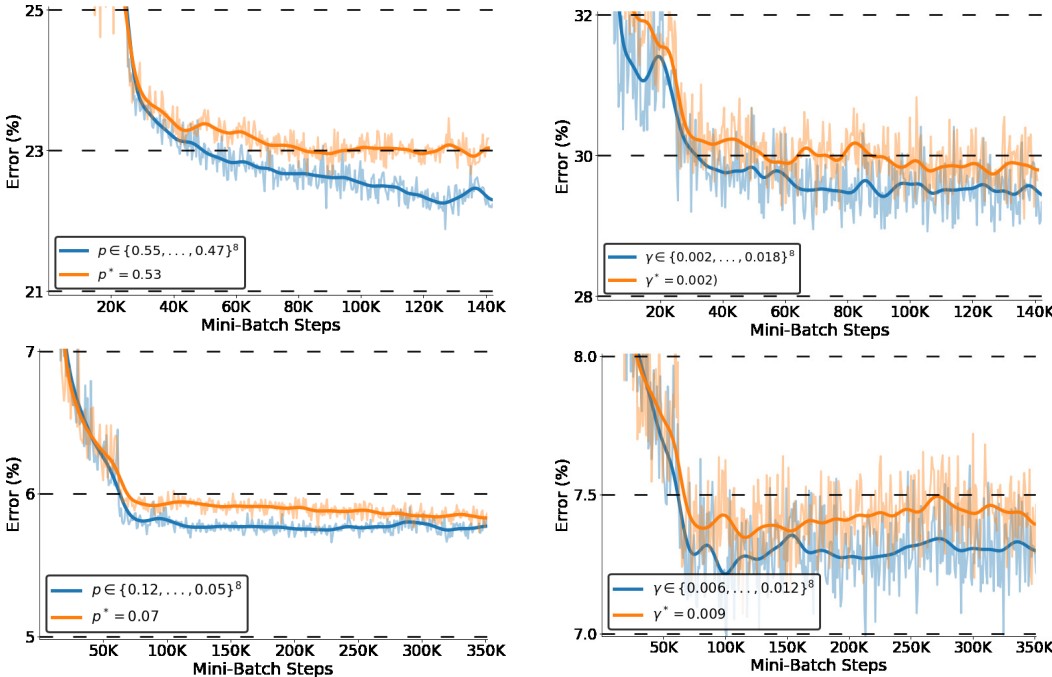

**Figure 3:** Test error curves for LeNet-like architecture for replica exchange training over Dropout $p$ and learning rate $\gamma$ . In the upper/lower row results for EMNIST / CIFAR-10 datasets. The dark lines are running averages.

Another input argument is the swap step; here an important observation is that when parallel tempering is used for MCMC sampling purposes, it is necessary to satisfy the detailed balance condition. For too small a step the system may not equilibrate sufficiently fast, and hence will not obey the Markov property.

## 4 EMPIRICAL RESULTS

To validate the proposed approach we conducted a series of experiments: first, using small LeNet-like models, we investigated the effects of various types of hyperparameter-induced "noise" on weight diffusion and tested the associated idea of hyperparameter replica exchange. We then moved to deep ResNet architectures He et al. (2016) to verify applicability of our parallel-tempering-based algorithm to large scale models.

The expriments were performed on EMNIST-letters (Cohen et al. (2017)) and CIFAR-10 (Krizhevsky & Hinton (2009)) datasets. EMNIST-letters consists of 124800 training images and 20800 testing images of shape $28 \times 28$ pixels with 26 classes. CIFAR-10 has 50000 training images and 10000 testing images with 10 classes. Each image has shape $32 \times 32$ pixels and 3 channels. The validation splits for both datasets were generated by random sampling of a training dataset taking 10% out. All models were trained with a mini-batch size of 128.

The LeNet-like models are summarized in tables 1 and 2. For EMNIST we apply zero-padding to $32 \times 32$ pixels before the inputs are fed to the network. Each average pooling layer is multiplied by a learnable coefficient and added bias (one per feature map). The neurons between the average pooling layer and convolutional layer are fully connected (every feature map from average pooling

**Table 1:** Architecture for EMNIST

| Layer | Size | Activation |
|---|---|---|
| Input | $32 \times 32 \times 1$ | - |
| Convolution | $28 \times 28 \times 6$ | tanh |
| Avg Pooling | $14 \times 14 \times 6$ | tanh |
| Convolution | $10 \times 10 \times 16$ | tanh |
| Avg Pooling | $5 \times 5 \times 16$ | tanh |
| Convolution | $1 \times 1 \times 120$ | tanh |
| Fully Connected | 84 | tanh |
| Fully Connected | 26 | RBF |

**Table 2:** Architecture for CIFAR-10

| Layer | Size | Activation |
|---|---|---|
| Input | $32 \times 32 \times 1$ | - |
| Convolution | $28 \times 28 \times 6$ | relu |
| Max Pooling | $14 \times 14 \times 6$ | - |
| Convolution | $10 \times 10 \times 16$ | relu |
| Max Pooling | $5 \times 5 \times 16$ | - |
| Convolution | $1 \times 1 \times 120$ | relu |
| Fully Connected | 84 | relu |
| Fully Connected | 10 | softmax |

is connected to every convolutional feature map), as opposed to the original LeNet architecture from LeCun et al. (1998). In the output layer each neuron outputs the square of the Euclidean distance between its input vector and its weight vector. Afterwards we apply normalized exponential function (RBF activation). For CIFAR-10 we used Max Pooling instead of Average Pooling and softmax activation on the output. Every simulation starts with a learning rate of 0.1. The learning rate for EMNIST is annealed after 62K steps, and for CIFAR-10 after 25K steps. When considering dropout as the hyperparameter being varied, the exchanges are introduced after 25K and 10K steps for EMNIST and CIFAR-10, respectively. For the simulation where the learning rate itself is being varied, each replica is annealed to a different value of the learning rate, and exchanges start being proposed right after annealing.

The results for LeNet-like models are presented in Fig. 3. Replicas differing by a "temperature" defined by Dropout and learning rate are generated, and the model performance (in terms of classification error) of the best independent replica, corresponding to a fixed hyperparameter value, is compared against a parallel tempering solution, where replica swaps are introduced. In all of the cases, for both datasets, the best parallel tempered path achieves a significantly lower error rate. We have also observed increased resilience to overfitting in our small-scale simulations, though this aspect requires more careful study. It is worth noting, that for the EMNIST task an error rate comparable to the best untempered results is achieved in a much smaller number of training epochs. Overall, the training is at least as fast as for the independent simulations.

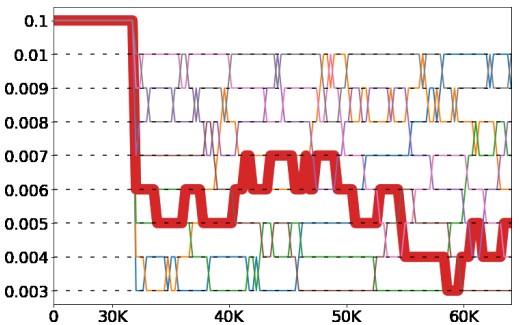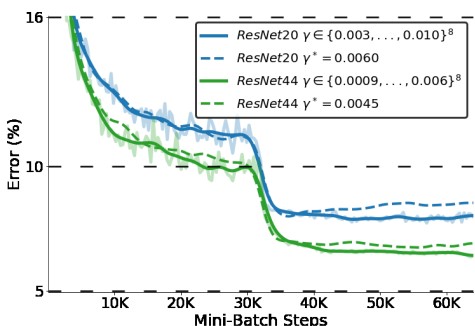

**Figure 4:** Left: Example of exchanges between replicas differing in learning rates, for the ResNet architecture. Shaded in grey: the path in hyperparameter space taken by the best simulation. Right: Test error curves for the ResNet architectures. Dashed lines correspond to the best annealed learning rate, while solid lines show the results with replica exchanges. The mean and standard deviations for multiple simulations are summarized in Table 3.

In order to showcase the flexibility of the algorithm, and its potential for generating additional improvements for already efficient and optimized models, we benchmark the approach on the residual architectures for CIFAR-10, where we follow He et al. (2016). Specifically, images are normalized by subtracting per-pixel mean and augmented via a number of transformations: the image is left-right-flipped with probability 0.5 and 4 pixels are zero-padded on each side, with a following random $32 \times 32$ crop. We use weight decay of 0.0001 and momentum of 0.9. We apply Batch Normalization, as in the original paper, and compare exchangeable learning rate to a fixed one after the initial learning rate annealing. The training begins with learning rate 0.1, annealed only once at step 32K; the total amount of training steps is 64K, as in He et al. (2016).

In Fig. 4 test error curves for ResNet20 and ResNet44 are presented, along with a visualisation of hyperparameter exchanges between replicas in a representative simulation. Introduction of the exchanges consistently reduces the overall minimal testing error for both architectures. With eight replicas we obtain an improvement of 1.04% for ResNet20 and 1.66% for ResNet44 (see table 3 for mean and standard deviation values). We expect larger improvements for larger number of replicas. Note that the hyperparameter value corresponding to the the best individually trained model (found by grid-search), is included in the parameter ladder for the PT simulation together with a small number of "suboptimal" values. The PT solution improves on all of them, by leveraging the replicas to explore the parameter space in a nonlocal fashion. This is seen in the non-trivial trajectories in hyperparameter space taken by all replicas as the training of their weights proceeds, in particular the trajectory of the ultimately best one.

**Table 3:** ResNet classification error for best independent and parallel-tempered (PT) solutions, from five runs with randomized 90/10 train-validation splits.

| Model | MEAN | STD |
|---|---|---|
| ResNet20 | 0.0791 | 0.0006 |
| ResNet20 PT | 0.0783 | 0.0011 |
| ResNet44 | 0.0674 | 0.0007 |
| ResNet44 PT | 0.0663 | 0.0014 |

## 5 CONCLUSIONS AND DISCUSSION

We introduced a new training approach to improve model optimization, by coupling previously independent grid search simulations at different hyperparameter values using the replica exchange technique. The method, which can be thought of as optimizing the model over non-monotonic and nonlocal paths in the joint hyperparameter/weight space, is very general: it can be applied to any hyperparameter which admits interpretation as a temperature-like quantity, in the very weak sense of facilitating weight diffusion during training. This diffusion test is, in fact, a simple experiment which can be performed at little cost to establish, whether any given parameter is related to landscape smoothness. We show that this is the case for Dropout, learning rate, but also Batch Normalization. The method is easily parallelizable, and similar in cost to standard grid search.

Experiments performed on LeNet (with CIFAR and EMNIST datasets) and ResNet (with CIFAR) architectures, showed consistently lower test error. In particular, we obtain improvement over the benchmark results for ResNet20 and ResNet44 on the CIFAR dataset.

A number of further improvements are possible. Practically, the most important concerns the automation of hyperparameter range selection, and the subsequent replica temperature ladder choice to optimize acceptances, which can potentially also reduce the test error. Conceptually, the idea can also be generalized to the case of multiple parameters controlling the smoothness of the energy landscape, which are not necessarily temperature-like Fukunishi et al. (2002). In this setup, multidimensional exchanges between replicas with different values of distinct hyperparameters are permitted, using the hyper-parallel tempering method Yan & de Pablo (1999); Sugita et al. (2000), allowing for more complex paths. These topics are the subject of an ongoing work.

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

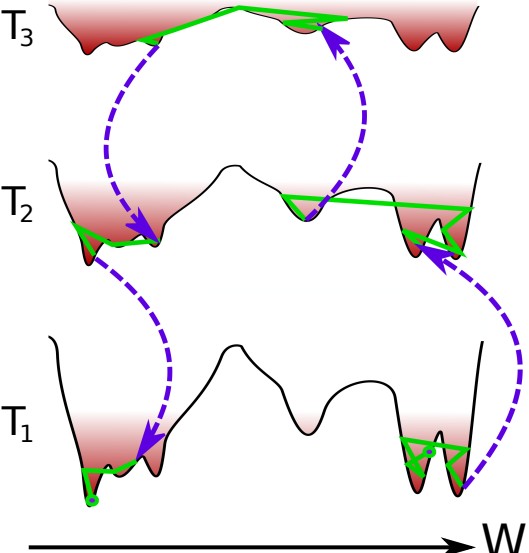

**Figure 5:** The training dynamics of the algorithm. The model replicas at different values of hyperparameters correspond to different fictitious temperatures $T_i$. At every temperature the algorithm performs *standard SGD optimization* (as opposed to Markov Chain Monte Carlo dynamics typically used in physical simulations), which is depicted with green trajectories. Periodically, swaps between neighbouring temperatures are proposed. The proposals are accepted using MC Metropolis-Hastings acceptance criteria, based on the current values of the loss functions $\mathcal{L}(W)$ and temperatures of the replicas [see Eq. 7]. Swap moves are shown with blue arrows in the figure. At higher temperatures the landscape is effectively flatter, and noisy SGD dynamics can access larger portions of it (the energy scale for SGD moves is shown with red shading). By using non-local moves mediated by higher temperature replicas the SGD dynamics can probe previously inaccessible regions of the potential landscape.

# A   APPENDIX

In Fig. 5 the dynamics of the training algorithm across multiple replicas is shown. We emphasize that the dynamics for any fixed value of a hyperparameter is performed using standard SGD dynamics, and not Monte Carlo. Moreover, SGD can be replaced by e.g. Adam optimization, as MC exchanges are only performed *between* replicas, within which dynamics can be arbitrary.

For conceptual simplicity we have assumed a single parameter $C$ controlling the acceptance ratio of swap moves between all the replicas in our proof-of-principle implementation. The performance of the algorithm is sensitive to the value of $C$. In the table 4 we provide a comparison of average acceptance rates for different values of $C$. In fact, for optimal performance acceptances between each pair of neighbouring replicas should be adjusted, to allow for an efficient exploration of the landscape at all scales. It may seem that this requires excessive tuning, in effect introducing additional hyperparameters, but this is not the case. The tuning of acceptances, or equivalently the distribution of hyperparameters for the replicas, can be performed automatically in an adaptive procedure Katzgraber et al. (2006); Trebst et al. (2006). In the current implementation we used a fixed acceptance parameter to focus the discussion on the main idea of replica exchange of hyperparameters.

**Table 4:** Comparison study of dependance of average acceptance rate across all the replicas on difference values of constant $C$.

| C | Average acceptance of move |
|---|---|
| 500 | 0.875 |
| 1000 | 0.785 |
| 5000 | 0.43 |
| 10000 | 0.304 |
| 50000 | 0.209 |

