# OpenReview forum: "Training Deep Neural Networks by optimizing over nonlocal paths in hyperparameter space"
_ICLR.cc/2020/Conference — Reject_

### Official Review · AnonReviewer2 · 2019-10-22
**Official Blind Review #2**

**Rating:** 6

**Review:**

This work presented an improvement of grid search algorithm for certain hyperparameters in deep neural nets training. These hyperparameters, such as learning rate and drop out, have "temperature" like meaning to control the noise injected in the training. With this analogy, the author proposed to use the idea of parallel tempering in statistical physics to allow exchange these hyperparameters during the training. Their empirical results showed this improves standard grid search.

The main contributions are two folds: 1. the connection between some hyperparameters to its physical perspective (temperature) and 2. the empirical validation of the proposed algorithm. The paper is clearly written and easy to follow. I would like to accept this paper if the authors can address following comments.

1. In Line 11 of algorithm 1, what is \alpha and how to update it?
2. How do the authors come up with the values for learning rate and drop out in the experiments? How sensitive the results are to the choice of those values?
3. What is the value of C used in the experiments. Do you have ablation study of this choice? How to pick this value?

**Experience Assessment:**

I have published one or two papers in this area.

**Review Assessment: Checking Correctness Of Derivations And Theory:**

N/A

**Review Assessment: Checking Correctness Of Experiments:**

I assessed the sensibility of the experiments.

**Review Assessment: Thoroughness In Paper Reading:**

I read the paper at least twice and used my best judgement in assessing the paper.

---

> ### Author Response · Authors · 2019-11-14
> **Response to Review 2**
>
> We thank the Referee for his comments and careful reading of the manuscript. We would like to address his remarks:
>
> 1. We thank the Referee for pointing out this typo. \alpha is a notation for the fraction of the accepted moves, which is used for book-keeping and for clarity was removed from the manuscript discussion. We corrected this error in the revised version.
>
> 2. We select dropout and learning rates in an equidistant manner in between the minimal and maximal value of the parameters. In this respect the window is selected based on diffusion experiments: at high “temperatures” the model exhibits a fast diffusion and (weak) convergence of objective function, at low temperatures the model tends to overfit. Distances between replicas are set such that there is a reasonably large acceptance rate for moves between replicas. We emphasize, however, that all this was done in our basic implementation, and as we remark below (and in Appendix in the revised manuscript), it can be fully automated -- hand-tuning of those quantities is not necessary.
> This, and the Referee’s last question about the constant C, touch upon the same important issue: in parallel tempering simulations the crucial quantities which do influence the results are the acceptances of swap moves between the replicas. Those acceptances guarantee that the simulation ergodically explores the landscape at all scales. In physical PT simulations, it is empirically known that the results are not sensitive to the detailed values of temperatures of individual replicas, as long as the acceptances between them are good and the overall parameter interval covered by all of them remains the same.
>
> 3. The question about parameter C and the replica hyperparameter values are linked. The acceptances are tuned by adjusting the separation between temperatures (hyperparameter values), as seen in Equation 7 in the manuscript. We tune this alternatively by adjusting parameter C, as in line 7 of the algorithm. We provide the numbers in the code that accompanies the paper. We show the sensitivity of this acceptance to the value of C in the table in the new appendix.
>
> In the state of the art replica exchange algorithms the acceptances, or multiplicative constants, are tuned automatically in an adaptive procedure during the run (we provide references in the Appendix of the revised manuscript).

---

### Official Review · AnonReviewer3 · 2019-10-25
**Official Blind Review #3**

**Rating:** 3

**Review:**

REVIEW 2

The paper proposes a new paradigm to perform hyperparameter search by proposing a way to jointly optimize over the hyperparameter space and the parameter space as opposed to the traditional way to performing these in separation (with the hyperparameter search invoking the parameter optimization) The paper’s main claim is that this allows the seach to follow non-local paths in the joint space. The main methodology proposed in the paper is inspired by the idea of parallel tempering from physics. The paper proposes to view parameter learning under a certain hyperparameter as running a langevin chain at a particular temperature. This is motivated by considering common hyperparameters as batch size, dropout rate or learning rate as inducing a specific level of noise to the training process, the variance of which is analogous to the inverse temperature in Langevin diffusion.

Under the above analogy, the paper proposes to run an analogue of parallel tempering in the following way. Consider running parallel threads of parameter optimization for various fixed choices of hyperparameter values (each representing a temperature). After running certain number of steps of SGD in each configuration, the paper proposes to exchange hyperparameters within a pair of threads, with the chance of exchange depending upon a criteria motivated by metropolis hastings.

While the paper proposes an interesting idea I have many reservations about the paper and applicability of the idea in practice prompting my score which I outline below.

1. The first and major shortcoming of the paper is an extreme lack of detail and rigour. The paper seems to be written in a very hand wavy fashion. While that is okay in working towards the proposal of the algorithm, which of course is merely claimed to be inspired by parallel tempering. I believe the paper needs to do an important job of specifying exactly how one selects the notion of temperature for each choice of the hyperparameter. The mapping of the choice to the temperature value is essential for the algorithm to succeed and as described in the paper, there is no hope of replicating the algorithm as the authors are extremely vague about how to figure out this mapping. A second instance of this is diffusion curves in the introduction. The paper does not define what a numerical value of diffusion is. It only relies on a hand-waved understanding of diffusion to even understand presented curves. Note that the paper does not have an appendix and there is no link to code either.

2. Lack of intuition - for this being a good procedure in terms of selecting a good path eventually. Note that hyperparameter optimization is an optimization procedure and it is very unclear to me why sampling from the distribution which will look like the product distribution over the replicas is even a good idea. If it is the case this needs to be specified carefully.
3. The experiments performed seem to follow a common trend. A single hyperparameter is selected (learning rate/dropout rate), a small grid of fixed values for the hyperparameter are selected and the procedure is performed. In light of this, what a “good” path through the algorithm represents is a good schedule for adjusting these hyperparameters. This is a well known intuition that for parameters like learning rate one needs to adjust them periodically to achieve good training dynamics. Since the comparison is to fixed learning rates, the improvement therefore looks predictable.

4. Lack of applicability of the algorithm for hyperparameter search in practice - There are a couple of things which make the algorithm restrictive -
This seems to be useful for hyperparameters that can be in a sense equated to parameters that change the training dynamics. The examples considered in the paper - batch size, learning rate, dropout rate. I think the authors should make this distinction and state it upfront because there are many hyperparameters that we optimize over which change the target distribution and not just the noise.
Secondly, large scale hyperparameter search takes the idea of separating hyper parameter and parameter search as then it can run in a parallel asynchronous fashion. The algorithm is highly synchronous way of doing this search. While it can be argued that we should move to this paradigm if the results are significant but unfortunately no comparisons with existing hyperparameter search algorithms are provided.


**Experience Assessment:**

I have read many papers in this area.

**Review Assessment: Checking Correctness Of Derivations And Theory:**

I carefully checked the derivations and theory.

**Review Assessment: Checking Correctness Of Experiments:**

I assessed the sensibility of the experiments.

**Review Assessment: Thoroughness In Paper Reading:**

I read the paper at least twice and used my best judgement in assessing the paper.

---

### Official Review · AnonReviewer1 · 2019-10-27
**Official Blind Review #1**

**Rating:** 3

**Review:**

This paper presents an analogous optimization algorithm that is inspired by MCMC. The algorithm maintains a series of weight replicas and sweeps them according to loss functions, based on rules from statistical physics.

I appreciate the idea of introducing temperature here, and the part that describes the idea is nicely written. I may have concerns, as in training neural network, most optimization algorithms use stochastic gradient which intrinsically contain noise inside, although certain level of gradient noise helps to escape saddle point on the landscape, there is no necessary analysis whether this additional noise would contribute to the optimization positively or negatively, after all the noise level has to be bounded for the optimizer to converge.

The experiments are a little insufficient, as the authors used very small datasets, and there is no comparison with other carefully tuned optimization algorithms, which, circle back to my previous concern, is that whether such an algorithm could actually outperform a carefully tuned SGD, ADAM.

**Experience Assessment:**

I have published one or two papers in this area.

**Review Assessment: Checking Correctness Of Derivations And Theory:**

I assessed the sensibility of the derivations and theory.

**Review Assessment: Checking Correctness Of Experiments:**

I assessed the sensibility of the experiments.

**Review Assessment: Thoroughness In Paper Reading:**

I read the paper at least twice and used my best judgement in assessing the paper.

---

> ### Author Response · Authors · 2019-11-14
> **Response to Review 1**
>
> We thank the Referee for the report and finding the main idea of our approach interesting. We would like to clarify a few misconceptions which are likely due to shortcomings of exposition. We added a short appendix to the paper, as well as a figure, to improve clarity.
>
> 1. The algorithm uses SGD dynamics within each replica, it is only in the stochastic swaps between the replicas where Monte Carlo transitions are attempted. In fact, our algorithm is a meta-approach, in that the dynamics within each replica can equally well be performed also using e.g. ADAM optimizer. In practice, a comparison between using different such dynamics would be valuable.
> 2. As the Referee correctly points out, the SGD dynamics inherently contains noise. We actually exploit that fact by using SGD within replicas, and we can do parallel tempering also on this noise, for instance on batch size (and we do parallel temper on the learning rate).
> 3. We do compare our algorithm with hyperparameter grid search + SGD, even for the small networks and datasets. Additionally, we perform a comparison with benchmark results for ResNet, which was a finely tuned and state-of-the-art result at the time of publishing. We show that our method improves on this result, even with our proof-of-principle implementation. We expect additional gains with carefully optimized production implementation and a larger number of replicas.

---

### Author Response · Authors · 2019-11-14
**Changes from original submission**

In the revised version of the manuscript, we edited the Algorithm section and removed reference to variable \alpha - which is a typo and is not used in the final version of the text for clarity. We also added a short appendix with additional figure and a table to clarify the training dynamics of the algorithm and the role of the constant C.

---

### Decision · Program_Chairs · 2019-12-19

**Decision:**

Reject

**Comment:**

This paper uses a variant of parallel tempering to tune the subset of neural net hyperparameters which control the amount of noise and/or rate of diffusion (e.g. learning rate, batch size). It's certainly an appealing idea to run multiple chains in parallel and periodically propose swaps between them. However, I'm not persuaded about the details. The argumentation in the paper is fairly informal, and it uses ideas from optimization and MCMC somewhat interchangeably. Since the individual chains aren't sampling from any known stationary distribution, it's not clear to me what MH-based swaps will achieve.

The authors are upset with one of the reviews and think it misrepresents their paper. However, I find myself agreeing with most of the reviewer's points. Furthermore, as a general principle, the availability of code doesn't by itself make a paper reproducible. One should be able to reproduce it without the code, and one shouldn't need to refer to the code for important details about the algorithm.

Another limitation (pointed out by various reviewers) is that there aren't any comparisons against prior work on hyperparameter optimization. Overall, I think there are some promising and appealing ideas in this submission, but it needs to be cleaned up before it's ready for publication at ICLR.